# Sustainability of Water Resources in Shandong Province Based on a System Dynamics Model of Water–Economy–Society for the Lower Yellow River

**Xue Zhang [1], Lirong Xu [1,*] and Chunhui Li [2,*]**

1    School of Water Conservancy and Environment, University of Jinan, Jinan 250024, China; zhangxue7080@163.com
2    School of Environment, Beijing Normal University, Beijing 100875, China
*    Correspondence: stu_xulr@ujn.edu.cn (L.X.); chunhuili@bnu.edu.cn (C.L.)

**Abstract:** The sustainability of water resources is a common issue for all human beings. In order to solve the water resource shortage problem in the lower Yellow River region of China, this paper uses a system dynamics approach from the perspective of a water–society–economy coupled model for the sustainable utilization of water resources in the lower Yellow River region in Shandong province. The model was validated using the water quantity, economic, and demographic statistics of nine prefecture-level cities in Shandong province from 2011 to 2020. Based on this model, three analysis scenarios were set up. The sustainability of water resources in the lower Yellow River region of Shandong province was analyzed by integrating and regulating indicators in the coupled model. The research shows that, with the liberalization of the population policy, water shortages will become critical. However, by appropriately reducing the water consumed for economic needs, water deficiency can be resolved. According to the forecast of scenario analysis, scenario 3 (the Sustainable development scenario) was chosen as the optimum solution. Assuming that the growth rate of agricultural and industrial production is controlled to reduce water consumption, the water deficiency rate of Shandong province will decrease year by year, and eventually the water shortage situation will gradually improve from 2022.

**Keywords:** sustainable development; water resources; system dynamic model; scenario analysis; Shandong province



## 1. Introduction

Serving as an ecological barrier and economic zone, the Yellow River basin plays an important role in China. At the forum for the ecological protection and high-quality development of the Yellow River basin, President Xi pointed out: "Protecting the Yellow River is a grand plan for the great rejuvenation and sustainable development of the Chinese nation." [1]. The Yellow River basin has an important part to play in winning the battle of pollution prevention and control, playing a very important role in China's economic and social development and ecological security. High-quality development requires higher standards to maintain the sustainable development of water resources in the Yellow River basin. However, due to limited natural resources and carrying capacity, the Yellow River basin has been facing problems of a fragile ecological environment and severe water resource security. Specifically, the development and utilization of water resources in the Yellow River is unreasonable. The unbalanced distribution of water resources in various regions leads to insufficient water supply in some regions. With the rapid growth of the population accompanied by industries and agriculture that need a lot of water resources, the contradiction between supply and consumption of water resources has increasingly intensified, being difficult to solve and seriously hindering the development of social economy. In addition, the traditional irrigation backward tool results in the waste of water

resources, which makes it difficult to use water resources effectively. The supply of the irrigation area is greater than the demand, although the downstream irrigation area is in urgent need of water resources. Coupled with natural and management factors, the waste of water resources in the Yellow River irrigation area is exacerbated.

To realize the sustainability of water resources, many scholars have conducted in-depth research on water resource utilization, water resource ecological protection, ecological footprint, and so on.

In terms of water resources, Dawadi used system dynamics to study the sustainable development of water resources. Combined with water conservation measures and water pricing, the impact of population growth and climate change on water volume in the Las Vegas Valley was studied [2]. Sušnik also established a system dynamics model to study the potential impact of water scarcity and socioeconomic policies on complex hydrological systems in the Kairouan region [3]. Yue Yusu constructed a system dynamics model for the beach area in the lower reaches of the Yellow River. He depicted the linkage relationship between water resource management, land resource allocation, and economic and social development of the beach area, and analyzed the impact of different magnitudes of floods on the economic and social system development of the beach area. In order to explore the operation law and change the trend of water resource supply and demand in Shiyan City [4], Huo Lei designed simulation schemes for water resource development and utilization under three scenarios based on the principles of system dynamics, and taking economic growth, population growth, and water supply sources into consideration to predict the future of the water supply and demand situation [5]. Qin Huanhuan put forward countermeasures to maintain the sustainable development of social economy and water resources in Longkou City after constructing the relationship between population, industry, agriculture, water resources and water environment [6]. Zhang Teng researched the supply and demand of water resources in the Haidian District, Beijing City, constructed a system dynamics model illustrating the balance between supply and demand of water resources, and provided countermeasures and suggestions for the development of the Haidian District after full consideration of population, social economy, and water resources [7].

In terms of ecological protection, Zhang Hongwu focused on the government of the Yellow River estuary and the protection of the wetland in the delta, which is of great significance for the ecological protection of the Yellow River and the high-quality development of social economy [8]. At the same time, Zhang also believes that it is necessary to consider the reality of the Yellow River basin to achieve benign governance of the lower reaches of the Yellow River [9]. Hao Fuqin analyzed the impact of the unified water regulation of the Yellow River on the downstream ecological environment from the perspective of the impact of water regulation on the downstream river ecosystem, the impact of downstream water body function, and the impact on wetland in the Yellow River delta [10].

In terms of ecological footprint, Li Zhongcai used the ecological footprint model to analyze the level of sustainable development of Shandong province from 2011 to 2020. It transpired that the ecological footprint of Shandong province showed a great increase, and the pressure on resources and the environment system gradually accelerated [11]. Luo Tao researched the supply and occupation of the ecological footprint of the existing flood control projects in the lower Yellow River, quantified the surplus level of ecological footprint, and proposed the guiding contents to optimize project planning, design, and construction and operation [12].

In terms of ecological water rights, the preliminary solution of the cut-off of the Yellow River is not equal to the realization of the ecological water rights of the Yellow River. Qiu Xiangwei, who believes that the institutional defect of ecological water rights is the fundamental method to solve the erosion of ecological water rights in the lower reaches of the Yellow River, explored the institutional root of the vulnerability of ecological water rights to erosion, and finally put forward several institutional optimization paths to protect ecological water rights [13].

Based on the above research, it can be concluded that water resources, social population, and economic indicators are the three key elements to determine the sustainability of water resources of the Yellow River, and there is a certain correlation between the three. However, at present, existing domestic research on this correlation is limited. It is necessary to quantitatively consider the systematic relationship of mutual influence and feedback among economy, society and policy factors that affect water demand, so as to capture the systematic behavior of water resource supply and demand. The system dynamics technique is used as a decision tool for engineering problems. It is one of the object-oriented approaches for studying and managing complex feedback systems [14].

The objective of this article is to provide reference to alleviate the shortage of water resources and to optimize the allocation of water resources in Shandong. Based on system dynamics, taking the lower reaches of the Yellow River in Shandong province as an example, we construct a model for the sustainable utilization of water resources by joining the perspectives water resources, society, and the economy. This is used to simulate the water resources coupled model of Shandong province from 2011 to 2020 and analyze the fitting degree affecting the water in Shandong province. On this basis, three development scenarios, maintaining the original development model and emphasizing the economic development model and the sustainable development model, are set up to simulate the sustainable situation of water resources in the lower reaches of the Yellow River in Shandong province over the next five years.

## 2. Materials and Methods

### 2.1. Study Area

Shandong province is located in the lowest reach of the Yellow River basin, with nine provinces belonging to the three major basins: Yellow River, Huai River and Hai River. The average annual water supply accounts for about 21% of the water consumption of the Yellow River, and for 1/3 of the total water consumption of Shandong province which is facing severe problems of water shortage and waste. The annual average amount of water resources in Shandong province is 30.3 billion cubic-meters, accounting for only 1.1% of the total amount of water resources in China, ranking in the last third across the country. At present, with the total water consumption exceeding the allocated distribution of the Yellow River, the resource competition between regional economic development and ecological environment preservation is fierce. Water shortage has limited regional high-quality development for a long time. In addition, the low reach where Shandong is located is the main producing area of agricultural products, and is a populated area in China.

This research includes nine cities scattered across Shandong along the Yellow River. The entire district location can be seen in Figure 1.

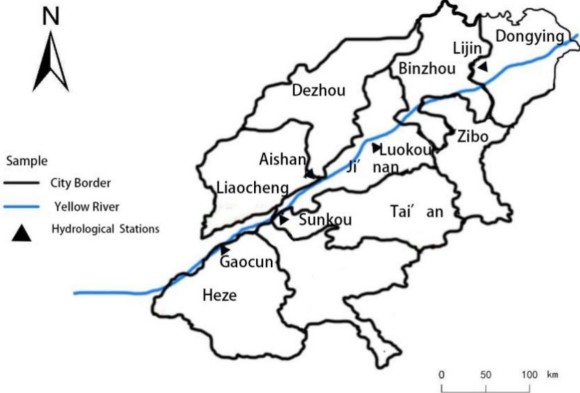

**Figure 1.** Research area of Shandong of the lower Yellow River basin.

Nine cities were divided into five areas by six hydraulic stations and considered as the samples. Districts division can be seen in Table 1. The center of Gaocun–Sunkou is Heze. The

center of Sunkou–Aishan is Ji'ning and Tai'an. The center of Aishan–Luokou is Ji'nan, Liaocheng and Dezhou. The heart of Luokou–Lijin is Zibo and Bingzhou. The heart of Lijin to the exit of the river is Dongying. J.T. in the row "Shorthand" refers to Jining–Tai'an, and so on.

**Table 1.** Division of regions in Shandong.

| District Region | City | Shorthand |
|---|---|---|
| Gaocun-Sunkou | Heze | H.Z. |
| Sunkou-Aishan | Ji'ning Tai'an | J.T. |
| Lookout | Ji'nan Liaocheng Dezhou | J.L.D. |
| Luokou-Lijin | Zibo Bingzhou | Z.B. |
| Lijin-Exit of River | Dongying | D.Y. |

*2.2. Method*

System dynamics models are guided and supported by qualitative analysis. This process requires that the model must be closely integrated with the actual situation; on the other hand, in-depth investigation and research must be conducted. As much information and statistical data as possible about system problems need to be collected. [15]

In this paper, the system dynamics method was used to construct a "water–economy–society" coupled model with the five regions of the lower reaches of the Yellow River in Shandong Province as the systematic research boundaries. Next, three subsystems, (water resources, economic, and social subsystems), were constructed through the system dynamics software STELLA V9.0.1. Then, variables formulas were put into the STELLA as well, defining and assigning values for each variable of the system, then running them.

*2.3. Model Construction*

Based on the water utilization condition in the lower Yellow River of Shandong Province, we established a coupled "water–economy–society" model. The model includes three subsystems: the water subsystem, the economy subsystem, and the society subsystem. In the water subsystem, the supplied water resources of Shandong are summed by annual average water, annual average surface supply, plus annual average (not duplicated with surface water) groundwater supply. Secondly, the economy system mainly includes consumed water resources, composed of agricultural water and industrial water. Lastly, the society subsystem demonstrates resident water consumption is significantly affected by the population of Shandong. The model structure is shown in Figure 2. All factors are explained in Table 2 [16].

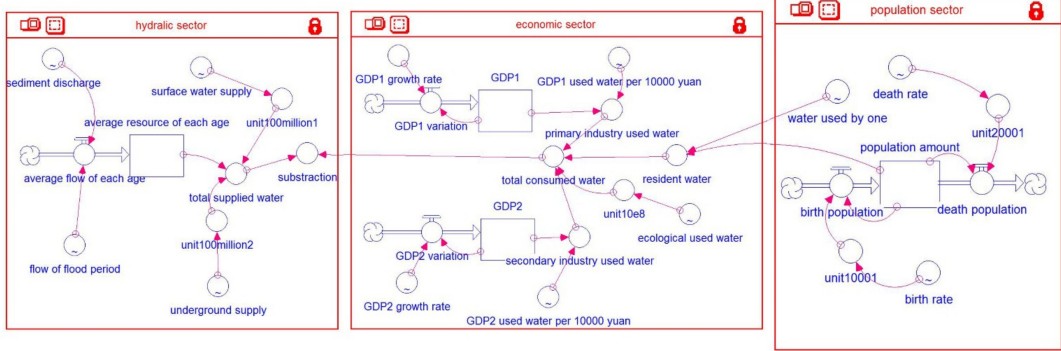

**Figure 2.** Model of Sustainable Water Resources Utilization System in the Yellow River Basin of Shandong Province.

**Table 2.** Explanations of all factors in three main subsystems [16].

| Subsystem | Variable Name | Explanation |
|---|---|---|
| Water | Total Supplied Water | Total water supply of public waterworks and social units that provide their own water sources |
| | Sediment Discharge | Mass of sand per cubic meter of water |
| | Flow of Flood Period | Average annual flow statistics from June to October |
| | Average Flow of each age | Average flow of each month in whole year |
| | Average Resource of each age | Average water resources of each month in whole year |
| | Surface Water Supply | Supply of crustal surface and water exposed to the atmosphere, in million cubic meters. |
| | Underground Supply | Water supply in saturated aquifers below the groundwater surface, in million cubic meters. |
| | Unit_100million_1 | Converting unit of surface water to standard |
| | Unit_100million_2 | Converting unit of underground water to standard |
| | Subtraction | Water deficiency: an indicator of water shortage |
| Economy | Total Consumed Water | Total amount of water used by water users in the area |
| | GDP1 Growth Rate | The ratio of the increase in the output value of the primary industry to the base period |
| | GDP1 Variation | The added value of the output value of the primary industry unit in a certain period |
| | GDP1 | Gross agricultural production at the end of year |
| | GDP1 Used Water for CNY 10,000 | Agricultural water consumption/10,000 yuan |
| | Primary Industry Used Water | Total water used for irrigation and rural livestock |
| | GDP2 Growth Rate | The ratio of the growth of the output value of the secondary industry to the base |
| | GDP2 Variation | The added value of the output value of the secondary industry per unit in a certain period |
| | GDP2 | Gross industrial production at the end of year |
| | GDP2 Used Water for CNY 10,000 | Industrial water consumption/10,000 yuan |
| | Secondary Industry Used Water | The total amount of production water used in the industrial production process and the domestic water used by employees in the factory area |
| | Resident Water | Total household water consumption |
| | Ecological Used Water | The minimum amount of water required for the restoration and construction of the ecological environment or to maintain the status of the current ecological environment |
| | Unit_10e8 | Converting ecological used water to standard |
| Society | Water used by one | Consumed water resources |
| | Population | Number of permanent residents at the end of the year |
| | Birth Population | Number of new-born residents |
| | Death Population | Number of life-loss residents |
| | Birth Rate | The ratio of birth population to average annual population |
| | Death Rate | The ratio of death population to average annual population |
| | Unit1_0001 | Converting birth rate into standard number |
| | Unit2_0001 | Converting death rate into standard number |

### 2.3.1. Water Subsystem

This subsystem is mainly used to calculate the total water supply, which is the sum of annual average water resources, annual average surface water supply and annual average (not duplicated with surface water) groundwater supply. It equates to:

$$\text{Total water supply} = \text{average annual water resources of the Yellow River} + \text{local surface water supply} + \text{local unduplicated groundwater supply} \quad (1)$$

Among them, the average annual surface water volume and average annual underground water volume were directly obtained from the historical data investigated [17]. The annual average water resources refer to the amount of foreign water from outside Shandong province, mainly from the Yellow River via Taohuayu to Lijin runoff, which can be deduced from the annual average flow. The average annual water resources are the accumulated water resources of the average annual flow in a year.

According to the studies of Lin Chen [18], it can be seen that the annual average flow $Q_{b5f}$ of the Yellow River basin in five years is correlated with the annual average sediment transport volume S and flood flow $Q_f$, which can be expressed as follows:

$$Q_{b5f} = k * S^\alpha * Q_f{}^\beta \tag{2}$$

After taking the logarithm of both sides of the equation, it can be expressed as:

$$lg(Q_{b5f}) = lg(k) + \alpha lg(S) + \beta lg(Q_f) \tag{3}$$

The corresponding coefficients can be solved by regression functions.

$Q_{b5f}$ represents the annual average flow in a certain area of the Yellow River basin in five years, in unit $m^3/s$. S represents the annual average sediment transport volume in the corresponding area, in unit $kg/m^3$. $Q_f$ represents the flood flow in each year in the region, in unit $m^3/s$.

$$\text{The average annual flow of an area} = k* \text{(sediment transport volume)} \,\hat{}\, \alpha * \text{(flood flow)} \,\hat{}\, \beta, \text{ where } k, \alpha, \beta \text{ are all coefficients} \tag{4}$$

$$\text{Average annual water resources of the Yellow River} = 86{,}400 * 365 * \text{average annual flow} \tag{5}$$

### 2.3.2. Economy Subsystem

In this subsystem, the economic subsystem is mainly used to calculate the total water consumption. The water consumption in the lower Yellow River basin of Shandong province mainly includes primary industry water consumption, secondary industry water consumption, ecological water consumption, and residential water consumption [17]. The most significant components are the value added of agricultural production and the value added of industrial production. The water consumption of the primary industry is determined by the added value of agricultural production and agricultural water consumption of CNY 10,000 [19]. Similarly, the water consumption of the secondary industry is determined by the added value of industrial production and industrial water consumption of CNY 10,000.

The equation in the model flow diagram is:

$$\text{Total water consumption} = \text{primary industry water consumption} + \text{secondary industry water consumption} + \text{ecological water consumption} + \text{residential water consumption} \tag{6}$$

$$\text{Gross agricultural value added} = \text{gross agricultural product} * \text{gross agricultural value growth rate} \tag{7}$$

$$\text{Gross industrial value added} = \text{gross industrial product} * \text{gross industrial value growth rate} \tag{8}$$

$$\text{Gross agricultural product (t)} = \text{gross agricultural product (t-dt)} + \text{gross agricultural value added} * dt \tag{9}$$

$$\text{Gross industrial product (t)} = \text{gross industrial product (t-dt)} + \text{gross industrial value added} * dt \tag{10}$$

$$\text{Primary industry water consumption} = \text{agricultural gross product} * \text{agricultural water consumption for CNY 10,000}/10{,}000 \tag{11}$$

$$\text{Secondary industry water consumption} = \text{industrial gross product} * \text{industrial water consumption for CNY 10,000}/10{,}000 \tag{12}$$

### 2.3.3. Society Subsystem

The social subsystem is used to calculate resident water consumption. Residential water consumption is determined by per capita water consumption and the total population.

The most important part of this subsystem is the total population amount. [15] The number of changes in the total population is caused by changes in the number of births and deaths. The number of births and deaths in each region is influenced by the birth and death rates.

The equation in the model flow diagram is:

$$\text{Residential water consumption} = \text{per capita water consumption} * \text{total population} \quad (13)$$

$$\text{Births} = \text{total population} * \text{birth rate} \quad (14)$$

$$\text{Deaths} = \text{total population} * \text{death rate} \quad (15)$$

$$\text{Total population (t)} = \text{total population (t-dt)} + (\text{births - deaths}) * dt \quad (16)$$

## 3. Model Validation

### 3.1. Parameters

The simulation period was from 2011 to 2020, with a time step in one year. The statistical data in this paper were mainly monthly flows from hydrological observation stations provided by the Yellow River Conservancy Commission over the years, including *The Yellow River Sediment Bulletin,* and *The Yellow River Water Resources Bulletin* from 2011 to 2020, the *Shandong Province Statistical Yearbook* provided by the Shandong Provincial Bureau of Statistics, and the *Shandong Province Water Resources Bulletin* provided by the Water Resources Department of Shandong Province. Details of sources are listed in Table 3.

**Table 3.** Sources of Parameters List.

| Subsystem | Parameters | Sources |
|---|---|---|
| Water Subsystem | Sediment transport volume<br>Flood flow<br>Local surface water supply<br>Local unduplicated groundwater supply | *Yellow River Sediment Bulletin*<br>*Yellow River Sediment Bulletin*<br>*Shandong Province Water Resources Bulletin*<br>*Shandong Province Water Resources Bulletin* |
| Economy Subsystem | Gross agricultural product *<br>Gross industrial product *<br>Agricultural water consumption for CNY 10,000<br>Industrial water consumption for CNY 10,000 | *Shandong Province Statistical Yearbook*<br>*Shandong Province Water Resources Bulletin*<br>*Shandong Province Water Resources Bulletin*<br>*Shandong Province Statistical Yearbook* |
| Society Subsystem | Total population<br>Per capita water consumption<br>Death rate<br>Birth rate | *Shandong Province Statistical Yearbook*<br>*Shandong Province Statistical Yearbook*<br>*Shandong Province Statistical Yearbook*<br>*Shandong Province Statistical Yearbook* |

Gross agricultural product *: The value of final agricultural products and services produced in a region's economy in a certain period, in CNY 10,000. Gross industrial product *: The value of final industrial products and services produced in a region's economy in a certain period, in CNY 10,000.

The model parameters can be classified as constants, table functions, and initial values of state variables. Parameters that are less time-dependent can be taken as constants, while parameters that are more time-dependent can be taken as table functions, which can accurately represent the nonlinear relationships between variables. In this article, most of the parameters are table functions. In this model, there were four initial values of state variables which were taken directly from the beginning of 2011, namely the annual average water resources, gross agricultural product, gross industrial product, and total population. Their main values are shown in Table 4.

**Table 4.** Initial Value of State Variables.

| District | Annual Average Flow/(m$^3$/s) | Gross Agricultural Product/CNY10$^4$ | Gross Industrial Product/CNY10$^4$ | Total Population/10$^4$ |
|---|---|---|---|---|
| H.Z. | 700.15 | 205.20 | 488.20 | 939.44 |
| J.T. | 655.44 | 440.73 | 2162.97 | 1387.14 |
| J.L.D. | 625.00 | 576.75 | 3053.51 | 1763.17 |
| Z.B. | 560.63 | 223.80 | 2299.18 | 798.91 |
| D.Y. | 462.20 | 74.73 | 1521.92 | 184.59 |

### 3.2. Validation Analysis

The validity and reliability of the system dynamics model can be demonstrated by comparing the relative error rates between the historical data and the simulated evolution data to verify the goodness of fit. The validation includes mainly gross agricultural product, gross industrial product, total population and annual average flow. According to Equations (1)–(14), these parameters can be conducted in STELLA. If the difference between the simulated value and the actual value is less than 10%, this model can be proved to be effective. Details of the comparison are shown in Figure 3. Coefficient values of annual average flow are shown in Table 5.

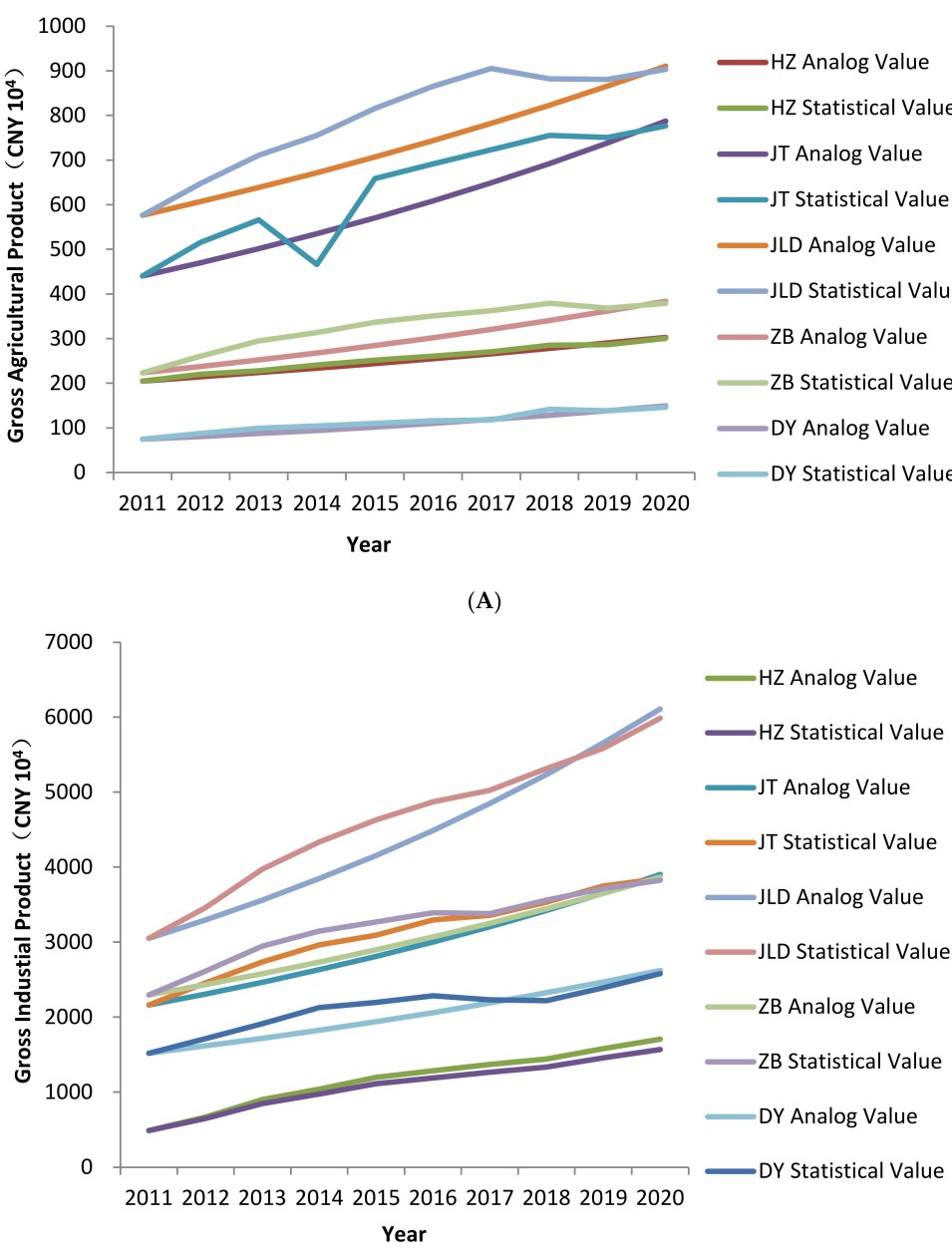

(**A**)

(**B**)

**Figure 3.** *Cont.*

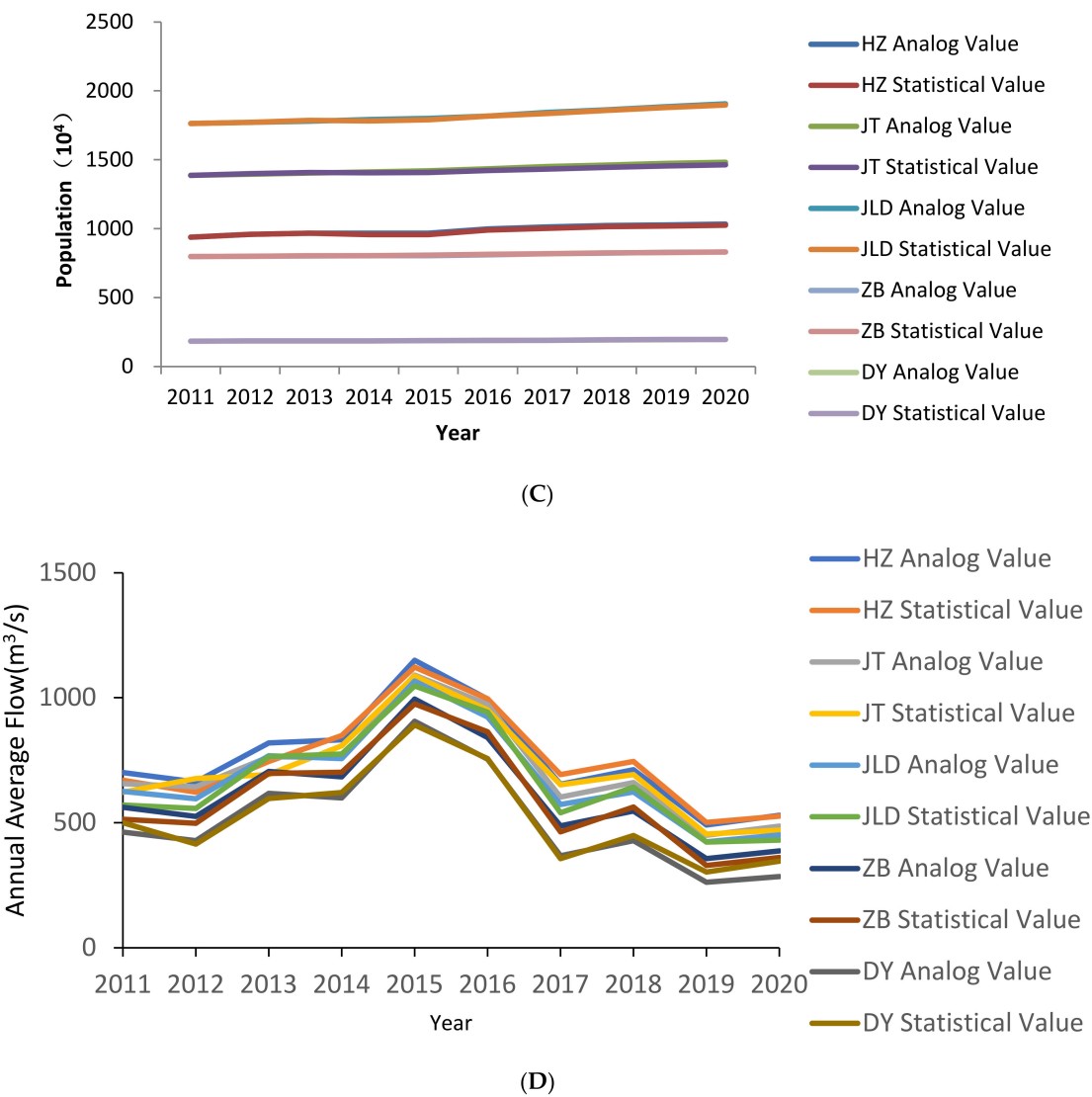

**Figure 3.** (**A**) Gross Agricultural Product (GAP) Simulated Results. (**B**) Gross Industrial Product (GIP) Simulated Results. (**C**) Population Simulated Results. (**D**) Annual Average Flow Simulated Results.

**Table 5.** Coefficient Value of Annual Average Flow Regression Equation.

| District | k | α | β |
|---|---|---|---|
| H.Z. | 126.94 | 0.090 | 0.240 |
| J.T. | 91.83 | 0.102 | 0.275 |
| J.L.D. | 66.07 | 0.118 | 0.308 |
| Z.B. | 55.98 | 0.137 | 0.312 |
| D.Y. | 21.73 | 0.177 | 0.415 |

According to the results, GAP, GIP, and population all showed an increasing trend year by year, indicating that the economic development of Shandong Province has gradually stabilized. The average annual flow increased first and then decreased in the last ten years, while in recent years the average annual flow tended towards vibration with little change. From Figure 3A, we can see that, except for J.T. and J.L.D., the simulated values of other regions were closer to the statistical value curve, with a good degree of fit. In Figure 3B, all regions had a good fit of gross industrial product and their simulated values had an average annual growth rate of 15.0%, 6.78%, 8.01%, 5.95%, and 6.24%, respectively. The simulated values of these regions in 2020 were 3.50 times, 1.80 times, 2.00 times, 1.68 times,

and 1.72 times of those in 2011. It can be seen from Figure 3C that the population growth rate in all regions was modest and maintained a low level of population growth. The average population growth rate of each region was 10.7‰, 7.45‰, 8.71‰, 4.33‰ and 6.70‰ respectively. The simulated values of these regions in 2020 were 1.48 times, 1.79 times, 1.58 times, 1.72 times and 2.00 times of those in 2011. According to Figure 3D, the simulated value of annual flow in all regions was close to the actual value, and the annual flow in each region showed a trend of first increasing and then decreasing. In 2015, the annual average flow reached the maximum, and in 2020, the annual average flow in each region remained within 200~600 m$^3$/s.

After comparing the data, the relative errors between the simulated values and the real values of the past years were maintained within 10%, indicating that the model results were basically consistent with the historical data, and the model could be considered to be consistent with the system in this study.

## 4. Scenario Analysis

### 4.1. Scenario Design

To study the sustainable utilization of water resources, this paper mainly concentrated on water supply and water consumption in the lower Yellow River of Shandong province. According to the results of model simulation from 2011 to 2020, it was proven that the coupled model offered value to the research on the sustainability of water resources of the Yellow River. Given this model, this section sets three scenarios. Three simulation scenarios with different circumstances expected from 2020 to 2025 were set up, as shown in Table 3, to seek the water shortage in the lower Yellow River in Shandong province in line with the future economic and social development trends. The adjusted amount can be found in Table 6.

Scenario 1 Basic Type

It is hypothesized that the law of development and the law of water volume change in Shandong province is not significant, and that there is no significant economic growth, which means that it keeps on developing as it is, and the natural population growth rate is taken as 5‰ according to the existing average growth rate in national statistics.

Scenario 2 Economic Development Type

The change mainly focuses on the importance of economic development, followed by the management of the riverbank. It is hypothesized that the agricultural GDP growth rate and industrial GDP growth rate are reduced by 2% and that the natural population growth rate is taken as 5‰.

Scenario 3 Sustainable Development Type

This scenario requires stable economic development, with a 2% increase in agricultural GDP growth rate and industrial GDP growth rate. Due to the improvement of production process and awareness of water conservation, the water consumption of agriculture and industry is set to decrease by 20% in this scenario, and the natural population growth rate is increased to 6‰ in combination with China's gradually liberalized population policy.

**Table 6.** Adjusting Parameter of 3 Scenarios.

| Parameter | Scenario 1 | Scenario 2 | Scenario 3 |
|---|---|---|---|
| Agricultural GDP growth rate | - | - | −2% |
| Industrial GDP growth rate | - | - | −2% |
| Water consumption of 10,000 yuan agricultural added value | - | −20% | −20% |
| Water consumption of 10,000 yuan industrial added value | - | −20% | −20% |
| natural population growth rate | 5‰ | 5‰ | 6‰ |

### 4.2. Analysis and Result

This section combines five researched districts as a single one to conclude water deficiency for the whole of Shandong. The statistical results are mainly analyzed from the

perspective of the total water supply and water consumption. The water deficiency rate is the ratio of the difference between water consumption and water supply to the water consumption. The equation of water deficiency rate [5] is:

Water deficiency rate = (Water consumption - Water supply)/Water consumption * 100%  (17)

Within a limited range (−100%~100%), the closer the value is to 0, the better the balance between water supply and water consumption in the region. When the value is negative, the water supply is greater than the water consumption, indicating that the supply is sufficient [20].

Firstly, the relevant data of five regions were weighted on average. Secondly, the parameters required in the three scenarios were assigned. In order to calculate the total water supply, the initial values of variables such as average annual surface water volume and average annual underground water volume were input. To calculate the total water consumption, the initial values of variables such as agricultural growth rate and industrial growth rate were input. Then, STELLA was used to run the calculation. Finally, the simulation results of water deficiency rate in Shandong Province were conducted according to Equation (15).

The results of water deficiency rate changes for each scenario are shown in Figure 4, and the three scenarios were simulated with the STELLA software. The values of population growth rate for the regulating parameter in Scenario 1 were based on the average population growth rate of the last ten years in Shandong statistics. Scenario 2 had a higher requirement based on the industrial and agricultural economy. Scenario 3 was a sustainable optimization in terms of industrial and agricultural water consumption and population growth.

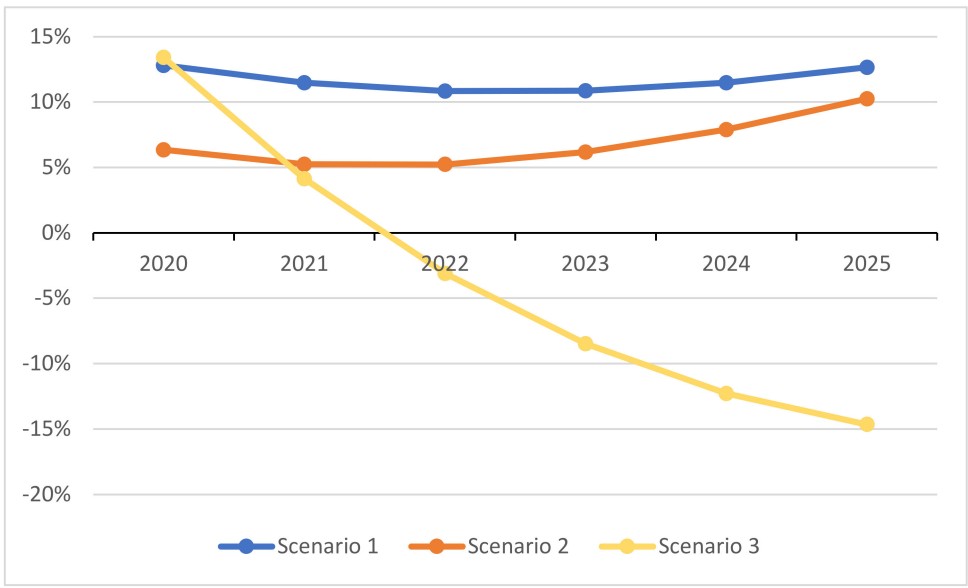

**Figure 4.** Water Deficiency in the lower Yellow River in Shandong Province.

In Scenario 1, as the population continues to grow, the water for domestic use will increase year by year, and the five regions will experience a continuous water shortage in the next five years.

Compared with Scenario 1, the overall water shortage is mitigated in Scenario 2 under the condition of lower water consumption for industrial and agricultural purposes, and the water deficiency rate will decrease noticeably by 2022, and from then on gradually gets out of control. This indicates that economic functions have a minor effect on water resources.

In Scenario 3, although the population increases year by year guided by policies, the average water consumption in industry and agriculture decrease as the level of energy-saving technology is updated. By 2025, the total water consumption in the lower Yellow River in

Shandong province will be 3.4 billion cubic meters and the total water supply will be 3.7 billion cubic meters. From 2022 onwards, the water deficiency rate will change to a negative value and the water shortage in Shandong province will be alleviated. The water deficiency rate in the lower Yellow River region decreases the most, reaching 30%. This illustrates that water consumption is the most critical element limiting sustainable development.

## 5. Conclusions and Discussion

### 5.1. Conclusions

This paper established a model for the sustainable utilization of water resources comprising water resources, social development, and economic development, based on the lower Yellow River in Shandong province with a system dynamics approach. Taking the data of Shandong province as an example, scenario analysis of water shortage in Shandong province for next five years was conducted and conclusions were found.

Firstly, this paper established a coupled "water–society–economy" system model of the lower Yellow River in Shandong province based on system dynamics using STELLA software. After collecting yearly data of five Shandong regions, it was verified that this coupled system model was available. It provides theoretical support for the study of the sustainable development of water resources.

Secondly, based on the establishment of model, the scenario analysis method was used to simulate sustainable utilization conditions for the next five years in Shandong. It was concluded that the overall water deficiency rate of Scenario 1 was larger than that of Scenario 2, and that of Scenario 2 was larger than that of Scenario 3, according to the scenario analysis. In Scenario 3, water consumption was basically sufficient to meet the water supply from 2022. By 2022, the water shortage in the lower Yellow River area of Shandong province would be improved through integrated regulation. The water deficiency rate of Scenarios 1 and 2 decreased by 2022, but water deficiency would recur by 2025, which needs to be further studied.

Thirdly, combining the main model and research results, reasonable suggestions were offered to avail water resources of Shandong. Integrated regulation of the social factors (natural population growth rate) and economic factors (water consumption of industrial and agricultural output) indicators of the coupled model would help to control the water resources in the lower Yellow River area of Shandong Province to effectively maintain a relatively stable state. This can ensure sufficient water resources in the lower Yellow River of Shandong province until 2025. The regulation of the water resources system should be enhanced in the future, and more attention should be paid to the conservation and reuse of water resources.

Thirdly, based on the main models and research results, reasonable suggestions on the utilization of water resources in Shandong province were offered. A comprehensive regulation of social factors (natural population growth rate) and economic factors (industrial and agricultural water consumption) in the coupled model would be helpful to effectively control water resources in the lower Yellow River region of Shandong Province to maintain a relatively stable state. Before 2025, the water resources in the lower reaches of the Yellow River in Shandong province could be sufficient. In the future, the regulation of water resource systems should be strengthened, and more attention should be paid to the conservation and reuse of water resources.

### 5.2. Discussion

In order to offer a comprehensive and specific study regarding the lower reaches of the Yellow River in Shandong province, it was necessary to analyze the water shortage of five sub regions, which are H.Z., J.T., J.L.D., Z.B. and L.J. in Shandong. Similarly, the three scenarios mentioned in Chapter 4 were used to simulate the changes of scenarios around the country and calculate the water shortage rate in the next five years under each scenario. Finally, the differences of water resource utilization in the five regions were compared.

Due to the complexity and uncertainty of system dynamics, there are some limitations in this study. At the same time, this paper needs further in-depth research. This model is affected by water resource, economic, and social subsystems. Particularly, water quality and reused water should also be considered. Water quality factors include whether each chemical factor meets the standard, which will directly affect the actual availability of water resources. The amount of reused water can not only affect the actual water demand of each region, but also affect the social awareness of water conservation.

**Author Contributions:** C.L. conceived and designed the research; X.Z. wrote the paper analyzed the data; L.X. contributed materials and analysis tools. All authors have read and agreed to the published version of the manuscript.

**Funding:** This study is supported by the National Key Research and Development Program (2018YFC0407403), and Shandong Provincial Natural Science Foundation (ZR2021ME145).

**Institutional Review Board Statement:** Not applicable.

**Informed Consent Statement:** Not applicable.

**Data Availability Statement:** Publicly available datasets were analyzed in this study. This data can be found here: (http://tjj.shandong.gov.cn/col/col6279/index.html), (http://wr.shandong.gov.cn/zwgk_319/fdzdgknr/tjsj/szygb/), (http://www.yrcc.gov.cn/zwzc/gzgb/gb/szygb/), (http://www.yrcc.gov.cn/zwzc/gzgb/gb/nsgb/).

**Acknowledgments:** We would like to extend special thanks to the editor and reviewers for insightful advice and comments on the manuscript.

**Conflicts of Interest:** The authors declare no conflict of interest.

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
