# Peer review of "Sustainability of Water Resources in Shandong Province Based on a System Dynamics Model of Water–Economy–Society for the Lower Yellow River"

_sustainability, doi:10.3390/su14063412_

Round 1
Reviewer 1 Report
In this article author established a model incorporating social factor and economic factors and exhibited that sustainable utilization of water resources reduces water consumption in industry and agriculture. The result and analysis are well presented. The spelling needs to be reviewed and the titles for figure 3 needs to be organized. If there is any other literature that shows the relationship between water resources, population, and economy, then please include it in the literature review.
Author Response
Point 1: In this article author established a model incorporating social factor and economic factors and exhibited that sustainable utilization of water resources reduces water consumption in industry and agriculture. The result and analysis are well presented. The spelling needs to be reviewed and the titles for figure 3 needs to be organized. If there is any other literature that shows the relationship between water resources, population, and economy, then please include it in the literature review.
Response 1: Thanks for your reviews.
1.After reconsidering my manuscript, most of spelling words have been rectified.
2.I have checked all Figure 3 and repositioned figures before. Now they should be matched as correct order from Figure 3A to D.
3.Some of references have been added, especially about relationship among water resources, population, and economy.
Reviewer 2 Report
-
The authors have covered a topic of “Sustainability of Water Resources Based on System Dynamic 2 Model of Water-Economy-Society in Shandong Province of 3 Lower Yellow River” which is an interesting study for the readers. Overall, the work is good and well written but there are some short comes in the manuscript which is needed to be revised before publishing this work.
- Carefully check large sentence structures, which are common throughout the MS.
- Main results or statistics findings of the model is missing in the abstract.
- “According to the forecast of Scenario analysis, the water shortage 22 would start to alleviate gradually by 2022”. How to claim this by the authors. please show main findings here.
- The main research gap and objectives of the study are missing in the introduction part.
- The citation style needs to check again. Some citations are missing.
- The method and data section need to be concise it’s too large.
- Results part needed minor revision.
- The discussion section of the paper is missing.
- Revised conclusion portion according to the results and conclude according to main findings of the study.
For more details about the comments and suggestions please check MS with comments.

Author Response
Point 1:Carefully check large sentence structures, which are common throughout the MS.
Response 1: Thanks for your reviews. The long sentence has been shortened.
Point 2: Main results or statistics findings of the model is missing in the abstract.
Response 2: Main findings are added.
Point 3: “According to the forecast of Scenario analysis, the water shortage 22 would start to alleviate gradually by 2022”. How to claim this by the authors. please show main findings here.
Response 3: In Abstract part, line. 22, an explanatory sentence is written to explain this opinion, “scenario 3 is chosen to be optimum solution. Assuming to control the growth rate of agricultural and industrial production and to reduce the water consumption, the water shortage rate of Shandong Province shall decrease year by year, and. Eventually the water shortage situation will gradually improve in 2022.”
Point 4:The main research gap and objectives of the study are missing in the introduction part.
Response 4: The main research gap and objectives of the study have been added in 1.Introduction part.
Point 5:The citation style needs to check again. Some citations are missing.
Response 5: Following the template format, all citations now are standardized.
Point 6:The method and data section need to be concise it’s too large.
Response 6: Thanks for your reviews. Following the researched model target, I have deleted complicated part and essential discription is left.
Point 7:Results part needed minor revision.
Response 7: Results part has been revised.
Point 8:The discussion section of the paper is missing.
Response 8: Discussion part has been added after “5.1 Conclusion”.
Point 9:Revised conclusion portion according to the results and conclude according to main findings of the study.
Response 9: Conclusion part has been more specific on the basis of researched process.
Point 10:For more details about the comments and suggestions please check MS with comments.
Response 10: I have viewed PDF file with comments and updated new manuscript.
Reviewer 3 Report
Introduction is focused on Yellow River, which is not bad. Present also cite references from abroad. Too few references.
Line 170. Even written literally, see template for equations.
Line 256-257. “With 256 computed in STELLA” word is missing?
Author Response
Point 1: Introduction is focused on Yellow River, which is not bad. Present also cite references from abroad. Too few references.
Response 1: Thanks for your reviews. Some of foreign references have been added.
Point 2:Line 170. Even written literally, see template for equations.
Response 2: All equations have been rewritten as template file mentions. The number order is listed behind the equations now, no more front.
Point 3: Line 256-257. “With 256 computed in STELLA” word is missing?
Response 3: In Line 256, ”With computed in STELLA, according to Equation 1 to 14, these parameters can be conducted.”, this sentence has been rewritten. “According to Equation 1 to 14, these parameters can be conducted in STELLA.”
Round 2
Reviewer 2 Report
Now it is publishing form.